# Impact of remotely sensed soil moisture and precipitation on soil moisture prediction in a data assimilation system with the JULES land surface model

Ewan Pinnington[1,2], Tristan Quaife[1,2], and Emily Black[1,3]

[1]Department of Meteorology, University of Reading, Reading, UK
[2]National Centre for Earth Observation, University of Reading, Reading, UK
[3]National Centre for Atmospheric Science, University of Reading, Reading, UK

*Correspondence to:* Ewan Pinnington (e.pinnington@reading.ac.uk)

**Abstract.** We show that satellite-derived estimates of shallow soil moisture can be used to calibrate a land-surface model at the regional scale in Ghana, using data assimilation techniques. The modified calibration significantly improves model estimation of soil moisture. Specifically, we find a 18% reduction in unbiased root-mean-squared differences in the north of Ghana and a 21% reduction in the south of Ghana for a 5-year hindcast after assimilating a single year of soil moisture observations to update model parameters. The use of an improved remotely-sensed rainfall dataset contributes to 6% of this reduction in deviation for northern Ghana and 10% for southern Ghana. Improved rainfall data has the greatest impact on model estimates during the seasonal wetting-up of soil, with the assimilation of remotely sensed soil moisture having greatest impact during drying down. In the north of Ghana we are able to recover improved estimates to soil texture after data assimilation. However, we are unable to do so for the south. The significant reduction in unbiased root-mean-squared difference we find after assimilating a single year of observations bodes well for the production of improved land surface model soil moisture estimates over sub-Saharan Africa.

## 1 Introduction

In regions where the population relies on subsistence farming it is soil moisture, rather than precipitation *per se* that is the critical factor in growing crops. The production of improved soil moisture forecasts should therefore enhance the drought resilience of these regions through improved capacity for early warning agricultural drought (Brown et al., 2017). Soil moisture is also an important variable for weather and climate prediction (Seneviratne et al., 2010), playing a key role in controlling land surface energy partitioning (Beljaars et al., 1996; Bateni and Entekhabi, 2012) and in the carbon cycle (McDowell, 2011). However, modelling soil moisture is complex and exhibits large sensitivities to meteorological forcing data and land surface model parameterisations (Pitman et al., 1999).

Globally, precipitation is the most influential meteorological driver in the estimation of soil moisture (Guo et al., 2006). However there is considerable variability in available precipitation data, which in turn has impacts on modelled predictions of soil moisture. When forcing a global land data assimilation system with different precipitation products Gottschalck et al. (2005) showed that the percentage difference in estimates of volumetric soil water content ranged between -75% to +100%. Similarly Liu et al. (2011) showed that driving a catchment land surface model with an improved precipitation product (merged gauge and satellite observations vs. a reanalysis product) increased the model soil moisture skill by 14%, when compared to in-situ observations.

There are now a variety of remotely sensed surface soil moisture observational products from both active and passive microwave sensors. Data assimilation (DA) has been used to combine information from these observations with land surface models to improve surface soil moisture estimates (Liu et al., 2011; De Lannoy and Reichle, 2016; Yang et al., 2016). DA refers to the suite of mathematical techniques used to combine models and observations combining available knowledge about their respective uncertainties. These techniques are typically derived from a Bayesian standpoint and can be broadly classified as sequential and variational. Sequential methods adjust the model state and/or parameters at the time when observations are available whereas variational methods adjust state and/or parameters at the beginning of some time window considering all observation within that window.

It has been shown by Bolten et al. (2010) that assimilation of remotely sensed surface soil moisture can significantly improve the prediction of root-zone soil moisture and drought modelling, where a sequential DA technique is used for soil moisture state estimation. Many other recent studies also use sequential assimilation methods to update the model soil moisture state at each time step when an observation is available (Liu et al., 2011; Draper et al., 2012; De Lannoy and Reichle, 2016; Kolassa et al., 2017). In addition some studies employing sequential methods estimate the model parameters as well as the state (Moradkhani et al., 2005; Qin et al., 2009; Montzka et al., 2011). Using sequential methods in this way will likely result in parameters that vary over time, which will not be optimal when using land surface models to run forecasts because the time-varying nature of the parameters will not be carried forward. An alternative is to use variational assimilation methods for parameter estimation (Navon, 1998). Variational methods will yield time-invariant parameter estimates over the assimilation time window. For a suitably chosen length of assimilation window (i.e. over one or more whole years) this allows us to avoid seasonally varying parameters. Using variational methods to assimilate remotely-sensed observations for land surface model parameter estimation has previously been shown to improve soil moisture estimates in several studies (Yang et al., 2007, 2009; Rasmy et al., 2011; Sawada and Koike, 2014; Yang et al., 2016). These studies all optimise both model parameters and state. Here we propose an alternative, which is to include the model spin-up within the data assimilation routine so that the initial soil moisture state is consistent with the updated parameters at each optimisation step.

The work in this paper forms part of the Enhancing Resilience to Agricultural Drought in Africa through Improved Communication of Seasonal Forecasts (ERADACS) project. Part of ERADACS is the development of a light-weight system for prediction of agricultural drought in Northern Ghana (TAMSAT-ALERT). Previous work (Brown et al., 2017) has shown that TAMSAT-ALERT's skill for predicting root-zone soil moisture in Ghana ensues largely from accurate knowledge of antecedent soil moisture conditions. In this paper we describe a method for improving soil moisture estimates for the Joint UK Land En-

vironment Simulator (JULES, see section 2.1) over Ghana through the assimilation of remotely-sensed soil moisture and use of improved satellite observed rainfall. Ultimately, we expect that the improved soil moisture estimates will increase the prediction skill of TAMSAT-ALERT, and hence the quality of drought early warning issued to farmers. We use the technique of Four-Dimensional Variational (4D-Var) data assimilation to estimate the soil thermal and hydraulic parameters of JULES by
assimilating European Space Agency Climate Change Initiative (ESA-CCI) merged active and passive microwave surface soil moisture observations (Dorigo et al., 2015). We also drive the JULES model with two successive versions of the TAMSAT rainfall dataset (see section 2.2) to investigate the effect of improved precipitation on soil moisture estimates. We assimilate a single year of soil moisture observations (2009), then perform a 5-year hindcast (2010-2014), driving the model with reanalysis meteorology, to judge the impact of both the precipitation products and data assimilation on the model's representation of soil
moisture when compared to independent observations.

## 2   Method

### 2.1   JULES land surface model

The Joint UK Land Environment Simulator (JULES) is a process based land surface model developed at the UK Met Office (Best et al., 2011; Clark et al., 2011). We used the global land configuration 4.0 of JULES designed for use across weather
and climate modelling timescales and systems (Walters et al., 2014). JULES is typically run with 4 soil layers, with the top layer being 10 cm deep. In this paper we have updated JULES to run with a top layer of 5 cm to be more representative of the ESA CCI soil moisture observations. Another option to deal with the issue of representativity would be an exponential filter (Albergel et al., 2008) which has been used in sequential data assimilation studies previously (Massari et al., 2015; Alvarez-Garreton et al., 2016). The model is forced with WFDEI data (WATCH Forcing Data methodology applied to ERA-
Interim reanalysis data), described by Weedon et al. (2014), for radiation, wind, temperature, pressure and humidity values. The WFDEI data has a $0.5°$ spatial resolution and a 3-hourly temporal resolution. The JULES model was run at a half-hourly timestep, with a soil map being taken from the harmonised world soil database (Nachtergaele et al., 2008). Previously JULES has been used in sequential DA experiments (Ghent et al., 2010), and has been implemented in a variational framework with focus on the carbon cycle (Raoult et al., 2016).

### 25   2.2   TAMSAT rainfall observations

We replaced the precipitation in the WFDEI data with Tropical Applications of Meteorology using SATellite data and ground-based observations (TAMSAT) rainfall monitoring products (Maidment et al., 2014; Tarnavsky et al., 2014). TAMSAT produces daily rainfall estimates over Africa at a 4 km resolution with data ranging back to 1983. The rainfall estimates are derived from Meteosat thermal infrared images calibrated against an extensive network of African rain gauges. When aggregated over time
and space, TAMSAT has been shown to have good skill over much of Africa, in comparison to ground-based observations (Maidment et al., 2013, 2017). On daily time scales, occurrence is better represented than amount (Greatrex et al., 2014), with

the magnitude of high intensity rainfall events not captured. For these reasons, TAMSAT tends to be used to monitor drought rather than to provide real-time early warning of floods. Data are available from https://www.tamsat.org.uk.

We ran JULES with WFDEI 3-hourly meteorological forcing data (Weedon et al., 2014) and TAMSAT daily rainfall estimates. Therefore we had to disaggregate the TAMSAT daily estimates to 3-hourly estimates. We did this by merging the TAMSAT data with the WFDEI precipitation data. We divided the WFDEI 3-hourly precipitation values by the corresponding WFDEI daily precipitation and then multiplied by the corresponding TAMSAT daily precipitation values. This spreads the daily TAMSAT estimates over the diurnal cycle.

In this study, we drive the JULES model with two different TAMSAT products (v2.0 and v3.0). The difference between JULES model outputs when forced with these two distinct products will help us to understand the impact of improved precipitation forcing on our estimation of soil moisture. TAMSAT v3.0 differs from TAMSAT v2.0 in that it uses an updated calibration against in-situ data that is more representative of local scales. It has been shown that TAMSAT v3.0 has greatly reduced the dry bias present in TAMSAT v2.0 (Maidment et al., 2017) and has eliminated the spatial artefacts. Despite this there are still areas where both products struggle, with coastal regions subject to large amounts of warm rain, and sharp topographic contrasts, being an example of this. For this reason, interannual rainfall variability is less well represented over the south of Ghana than the North. For more information on the differences between the two TAMSAT products see Maidment et al. (2017). In Figure 1 we show yearly cumulative rainfall averaged over 2009 - 2014 for TAMSAT v2.0 and v3.0, we can see the different spatial distributions of rain with v3.0 being wetter in the south and v2.0 wetter in the east. To illustrate the difference in the amount of rainfall for the two products we show cumulative rainfall for the period 2009 - 2014 averaged spatially over Ghana in Figure 2. This shows TAMSAT v2.0 to be the drier of the two products, as expected.

## 2.3   ESA CCI soil moisture observations

In this study we use the ESA CCI level 3 version 03.2 combined active and passive soil moisture observations. This product merges data from 11 different sensors, using an algorithm described in Dorigo et al. (2017) to give an estimate of surface soil moisture together with its associated uncertainty. These estimates are assumed to represent the top 2-5 cm of soil. However, observations based on different microwave frequencies and soil moisture conditions may be representative of deeper layers (Ulaby et al., 1982). It has been previously shown that it is best to use both active and passive retrievals together (Draper et al., 2012) and that the ESA CCI merged product performs better than either the active or passive product alone (Dorigo et al., 2015). Dorigo et al. (2015) also show that the ESA CCI product performs well over Western Africa when judged against in-situ soil moisture observations from the AMMA network (Cappelaere et al., 2009), with stations in Benin, Mali and Niger. When judged against the AMMA network CCI soil moisture was shown to have a high correlation ($\sim 0.7$) and one of the lowest unbiased root-mean squared differences ($\sim 0.04$) of the 28 worldwide networks used in the study. This bodes well for our comparison over Ghana, which has a similar climate regime in the north to the sites in the AMMA network. Figure 3 shows the number of available daily soil moisture observations in the experiment period (2009-2014) over Ghana, with the maximum number of possible observations being 2190. We can see that there is higher data availability in the north of Ghana than in the south. There are some pixels in the south for which we have no data, this is due to high vegetation cover.

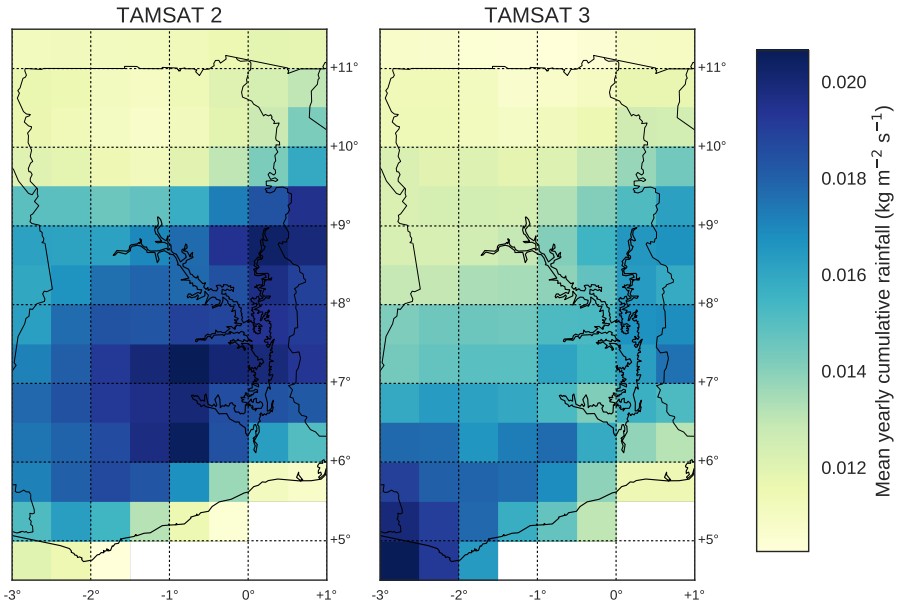

**Figure 1.** TAMSATv2.0 and v3.0 yearly cumulative rainfall averaged over the 6 years in our experiments (2009 - 2014).

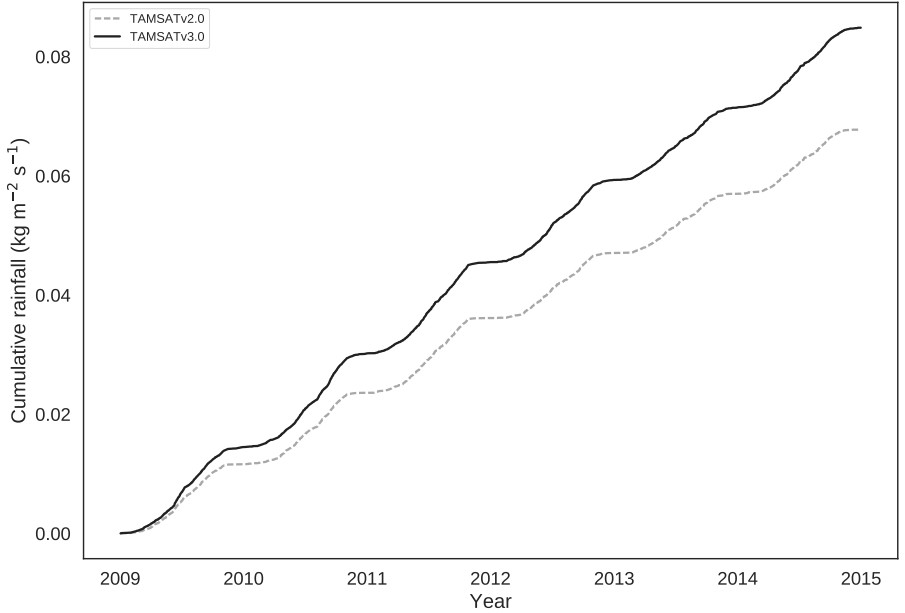

**Figure 2.** TAMSATv2.0 and v3.0 cumulative rainfall averaged over the whole of Ghana.

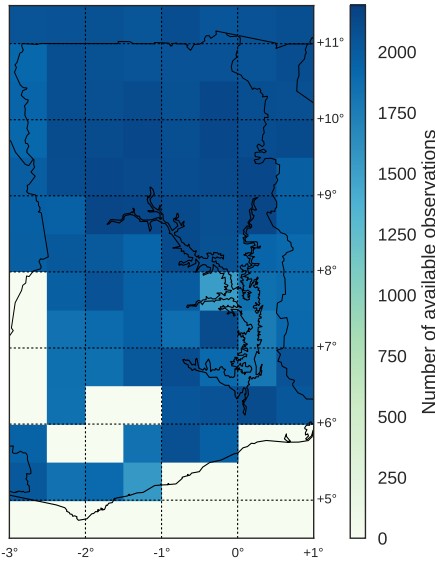

**Figure 3.** Number of available days of ESA CCI soil moisture observations in the experiment period (2009 - 2014) out of a maximum of 2190 days.

## 2.4    4D-Variational data assimilation

We use the method of Four-Dimensional Variational data assimilation (4D-Var) to estimate the soil thermal and hydraulic parameters of the JULES land surface model for each grid cell over Ghana. 4D-Var aims to find the initial state that minimises the weighted least squares distance to the prior guess while minimising the weighted least squares distance of the model
trajectory to the observations over the time window. This is done by minimising a cost function at each grid cell,

$$J(\mathbf{x}_0) = \frac{1}{2}(\mathbf{x}_0 - \mathbf{x}_b)^T \mathbf{B}^{-1}(\mathbf{x}_0 - \mathbf{x}_b) + \frac{1}{2}\sum_{i=0}^{N}(\mathbf{y}_i - \mathbf{h}_i(\mathbf{x}_0))^T \mathbf{R}_{i,i}^{-1}(\mathbf{y}_i - \mathbf{h}_i(\mathbf{x}_0)), \tag{1}$$

where $\mathbf{x}$ is the vector of model parameters, with $\mathbf{x}_b$ being a prior guess and $\mathbf{x}_0$ the current update, $\mathbf{B}$ is the prior error covariance matrix, $\mathbf{y}_i$ is the observation at time $t_i$, $\mathbf{h}_i$ is the observation operator (here the JULES model) mapping the model parameters ($\mathbf{x}_0$) to the observation $\mathbf{y}_i$ at time $t_i$, $\mathbf{R}_i$ is the observation error covariance matrix and $N$ is the number of observations. We
chose a variational DA method for parameter estimation over a sequential method, because variational methods ensure that the retrieved model parameters are time invariant over the assimilation window and will hence fit seasonal model dynamics when the window is sufficiently large. As we do not have a good estimate to the error in the prior estimates of model parameters we chose a conservative 5% standard deviation for the prior error covariance matrix $\mathbf{B}$. This ensures we do not retrieve unrealistic estimates of soil texture after data assimilation. For the observational error covariance matrix $\mathbf{R}$ we have a diagonal matrix with
variances estimated from the standard deviations included in the ESA CCI soil moisture product.

In this study, we updated the percentage of sand and silt in the soil at each minimisation step (with clay being updated implicitly) and then used a set of pedo-transfer functions (Cosby et al., 1984) to relate the new sand, silt and clay proportions to the 8 soil parameters in JULES. This is a similar framework to that introduced in Yang et al. (2007, 2009) for data assimilation with the Simplified Biosphere model 2 (SiB2) (Sellers et al., 1996), with the exception that the soil porosity parameter of JULES

is updated implicitly within the pedo-transfer functions rather than explicitly included in the optimisation. Parameterising the model in this way reduces the issue of equifinality (which potentially arises from minimising 8 related parameters) and decreases the convergence time of the minimisation. Our prior guess for the sand, silt and clay values at each grid cell comes from the harmonised world soil database. At each minimisation step after updating the parameters of JULES we included a model spin-up to ensure that the initial soil moisture state is consistent with the updated parameters. We used the Nelder-Mead

simplex algorithm (Nelder and Mead, 1965) to minimise the cost function in equation (1) without the use of a model adjoint. Whilst an adjoint facilitates efficient calculation of gradients in the cost-function it is costly to maintain and keep up–to–date with the latest model version. The only example of an adjoint of JULES for which we are aware is provided by Raoult et al. (2016) and is implemented for version 2.2 of the model, several major versions behind the current release. In future work a 4D-Ensemble-Var (Liu et al., 2008, 2009) approach could prove a useful compromise as it allows for the use of a gradient

based descent algorithm, reducing the total number of function calls required to reach a solution without the use of an adjoint.

## 2.5   Experimental design

For each data assimilation experiment with JULES (driven with TAMSAT v2.0 or v3.0 rainfall) we assimilate a single year of ESA CCI soil moisture observations (2009) and then run a 5-year hindcast (2010-2014). The hindcast allows us to evaluate the performance of each experiment against independent soil moisture observations. In our results, we consider 4 different model

runs:

1. JULES model "free-run" driven with TAMSAT v2.0 rainfall ("prior")

2. JULES model after calibration with DA, driven with TAMSAT v2.0 rainfall ("posterior")

3. JULES model "free-run" driven with TAMSAT v3.0 rainfall ("prior")

4. JULES model after calibration with DA, driven with TAMSAT v3.0 rainfall ("posterior")

From these 4 distinct experiments we can interrogate the impact of both the DA and use of the updated rainfall product.

## 3   Results

We split our analysis over northern and southern Ghana (above and below 9° N respectively) due to the issues of data quality between the two regions. The data quality of both precipitation and soil moisture is higher in the north than the south and also much of the subsistence agriculture in Ghana takes place in the northern regions, with a higher percentage of cash crops grown

in the south (Martey et al., 2013). In Figure 4 we show the results of a data assimilation and forecast for a single grid cell in the

north of Ghana, here both the prior (light grey line) and posterior (dark grey line) are forced with TAMSAT v3.0 precipitation (experiments 3 and 4 respectively, described in section 2.5). From Figure 4 we can see that the data assimilation has greatly improved the fit to the observations in the assimilation window (2009), which is to be expected, since these observations are what the model is calibrated against. However, the improved fit continues into the forecast (2010-2014) when comparing

against the unassimilated observations. We can see a distinct seasonal pattern for soil moisture in northern Ghana, where there exists a rainy season and corresponding "wetting-up" of soil moisture from approximately March-May and a dry season with "drying-down" of soil moisture from approximately November-January. The model skill for predicting this seasonal cycle is markedly improved after data assimilation, with a root-mean squared difference (RMSD) of 0.035 after data assimilation compared to a RMSD of 0.094 before, for 2009. In Figure 4 we can also see the amplitude of this seasonal cycle slightly

decreasing, this is a pattern also seen in both TAMSAT products which exhibit a drying over the period 2010-2014 for this grid cell. In Figure 5 we show the same model runs for a grid cell in the south of Ghana. The season in the south of Ghana is much less pronounced and this is seen in both the model runs and the observations. However, the observations are of poorer quality in the south due to the higher vegetation cover and cloud cover, adding to the noise seen in Figure 5. Although we do improve the fit to the observations after data assimilation in Figure 5 (RMSD of 0.059 after data assimilation compared to RMSD of

0.102 before, for 2009) we do not see the same scale of improvement as for the northern grid cell in Figure 4. This is most likely due primarily to the higher error in both the precipitation and soil moisture observations. In addition, the less pronounced seasonal cycle is more difficult to forecast after just assimilating a single year of data. The lower layer soil moisture in JULES responds in a similar way to the top layers shown in Figure 4 and 5, becoming slightly dried than our prior estimates after data assimilation. Without independent observations of these deeper layers it is difficult to know if this is realistic or not.

Figures 4 and 5 show results from experiments 3 and 4 when forcing the JULES model with TAMSAT v3.0 rainfall. In Figure 6 we show model Mean Relative Error (MRE) (judged against ESA CCI observations in the forecast period, 2010-2014 and calculated as the mean absolute deviation) for wet and dry seasons and experiments 1 to 4. Without DA (top row) we can see that for both wet and dry seasons there is a larger dry MRE in soil moisture in northern Ghana for TAMSAT v2.0 than v3.0 and a larger wet MRE in southern Ghana for TAMSAT v3.0 than v2.0. This finding is consistent with the comparisons

of precipitation between v3.0 and v2.0 presented by Maidment et al. (2017), where TAMSATv3.0 was shown to reduce a dry bias present in TAMSATv2.0 when compared to ground station data. After DA (bottom row) we can see that the wet MRE in southern Ghana is largely reduced for both TAMSAT v2.0 and v3.0. However, in northern Ghana a dry MRE still remains, with this being slightly drier for TAMSAT v2.0, compared to v3.0.

Figure 7 and 8 show experiment monthly root-mean-square differences (RMSD) for north and south Ghana respectively.

For Figure 7 this shows that the most accurate model run overall is experiment 4 (TAMSAT v3.0 with DA). We see in the majority of years that towards the start of the season as soils are wetting up it is experiment 3 and 4 (TAMSAT v3.0 no DA and with DA respectively) that have the lowest RMSD, suggesting that it is precipitation, as opposed to the assimilation of soil moisture, that is most important for improving soil moisture estimates during this period. This relationship changes towards the end of the rainy season with experiment 2 and 4 being the most accurate (TAMSAT v2.0 with DA and TAMSAT v3.0 with

DA respectively) suggesting that assimilation of soil moisture estimates is most important in this period. In Figure 8 the most

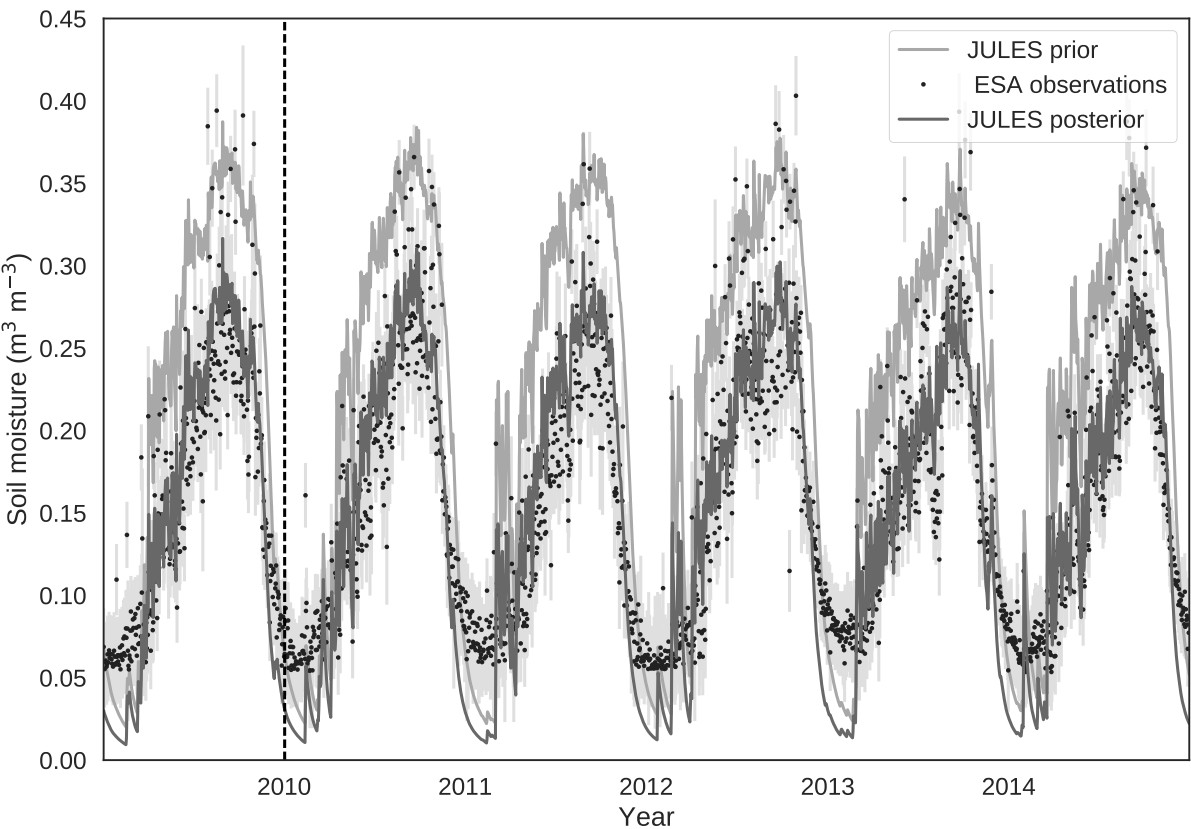

**Figure 4.** Soil moisture data assimilation results for a north Ghana grid cell using TAMSAT v3.0 driving data. Light grey line: prior JULES trajectory. Dark grey line: posterior JULES trajectory. Black dots: ESA CCI level 3 soil moisture observations. Faint grey vertical lines: error bars for observations. Vertical dashed line represents the end of the assimilation window.

accurate model run is again experiment 4 (TAMSAT v3.0 with DA), although experiment 2 (TAMSAT v2.0 with DA) is much closer in accuracy than for the north. This suggests that both rainfall products are poor in the south compared to the north. We also note that experiment 1 (TAMSAT v2.0 no DA) is markedly more accurate than experiment 3 (TAMSAT v3.0 no DA) in the south. However, considering the results after DA (experiment 4 outperforming experiment 2) this can be explained by an

5   incorrect specification of the prior soil map in the south rather than TAMSAT v2.0 rainfall outperforming TAMSAT v3.0 (it is expected that both products perform poorly in coastal regions (Maidment et al., 2017)). Experiments 2 and 4 have a lower RMSD in the south (Figure 8) compared to the north (Figure 7), this seems surprising given that we consider the quality of the data to be poorer in the south. However, this is in part due to the much more pronounced seasonal cycle in the north leading to peaks in RMSD when the seasonal cycle is even slightly mistimed by the model. We also have less confidence in the CCI soil

10   moisture observations in the south so a lower RMSD in comparison to this product over this region is perhaps not indicative of a better soil moisture estimate overall.

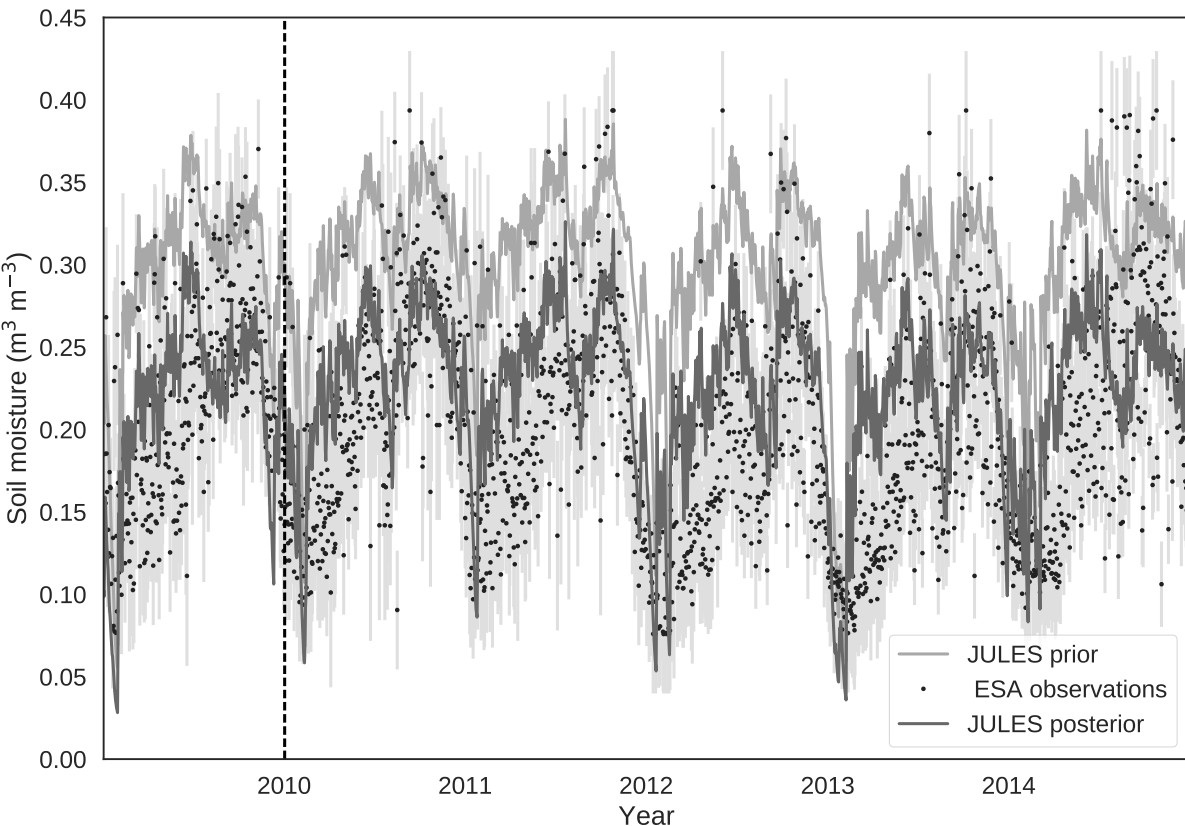

**Figure 5.** As Figure 4, except for southern Ghana.

Figure 9 compares the prior soil map used as the initial guess in the DA (i.e. from the Harmonised World Soil Data Base) with the posterior soil map retrieved by DA. The posterior soil map shown is the soil map retrieved when forcing JULES with TAMSAT v3.0 rainfall. It can be seen that after DA, the percentage clay is greatly reduced with increased percentages in silt and sand for the majority of grid cells. This change is reasonable for some grid cells, particularly in northern Ghana where soils

5  are often much more sandy/silty in texture (Braimoh and Vlek, 2004). Comparing estimates of soil texture derived from CCI soil moisture to in-situ observations is inevitably problematic due to issues of representativity in the spatial domain. However, independent sources of verification are difficult to find over Ghana. We therefore compare our soil maps to in-situ observations from The African Soil Profile Database (Leenaars et al., 2014). This database is compiled by the International Soil Reference and Information Centre (ISRIC) with the quality of the data being rated from 1 (highest quality) to 4 (lowest quality), here we

10  compare only to observations with a quality flag of 1 or 2. In table 1 we show the root-mean-squared error (RMSE) for our soil maps when compared to 21 in-situ observations of soil texture in the north of Ghana and 36 in-situ observations in the south (locations shown as red dots in Figure 9). For the north of Ghana where we have most confidence in our results we find a reduction in RMSE for both sand and clay (almost halving the RMSE in clay). However, the RMSE for silt is increased. In the

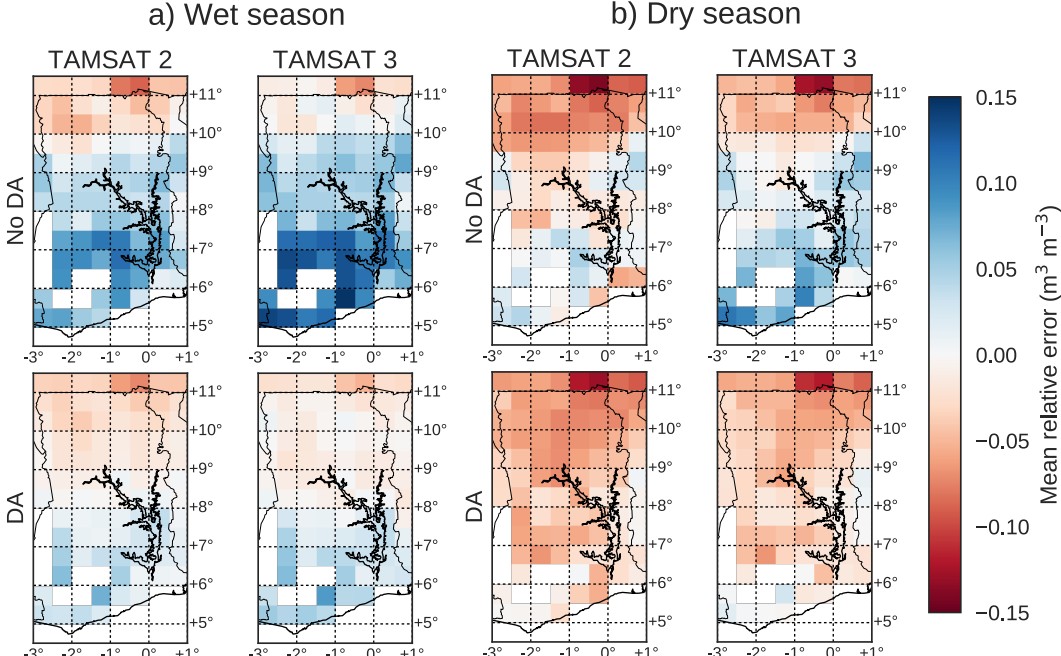

**Figure 6.** Soil moisture model minus observations for 5 year JULES forecast (2010-2014) driven with TAMSAT v2.0 and v3.0 precipitation and before and after data assimilation. Subplot a) statistics calculated over March to May for wet period, subplot b) statistics calculated over November to January for dry period. White pixels indicate areas where there is no data to calculate statistics (mainly due to high vegetation cover in the south).

south of Ghana we do not manage to recover a better estimate of soil texture after data assimilation, with an increase in RMSE for silt and clay but a decrease in RMSE for sand. The inability of the data assimilation to improve soil texture estimates at certain points is most likely due to issues of spatial representativity between the modelled soil map and the in-situ data. It is also possibly impacted by errors in our pedo-transfer functions, which may perform better if specifically calibrated for Ghanaian

5    soils (Patil and Singh, 2016).

Satellite soil moisture products can be subject to larger errors and biases associated with data processing. This is particularly true for the CCI level 3 combined active and passive product used in this paper, as in order to merge information from 11 different sensors data is cumulative distribution function matched to the GLDAS-Noah v1 model (Rodell et al., 2004). Therefore, any bias within the GLDAS-Noah model will be included in the level 3 soil moisture product used here. To make sure we are

10    not just correcting the bias of the JULES model to that of GLDAS-Noah we include summary statistics of unbiased root-mean squared difference (ubRMSD) and temporal correlation in table 2. In every case we find that after data assimilation we improve both ubRMSD and correlation and in the majority of cases find the best results for experiment 4 (TAMSAT v3.0 with DA). For the north of Ghana, we reduce the ubRMSD by 18% from experiment 3 ($0.0622$ $\text{m}^3$ $\text{m}^{-3}$) to experiment 4 ($0.0508$ $\text{m}^3$ $\text{m}^{-3}$). From experiment 2 to 4 we can see that, after data assimilation, using TAMSAT v3.0 rainfall over v2.0 has contributed to a

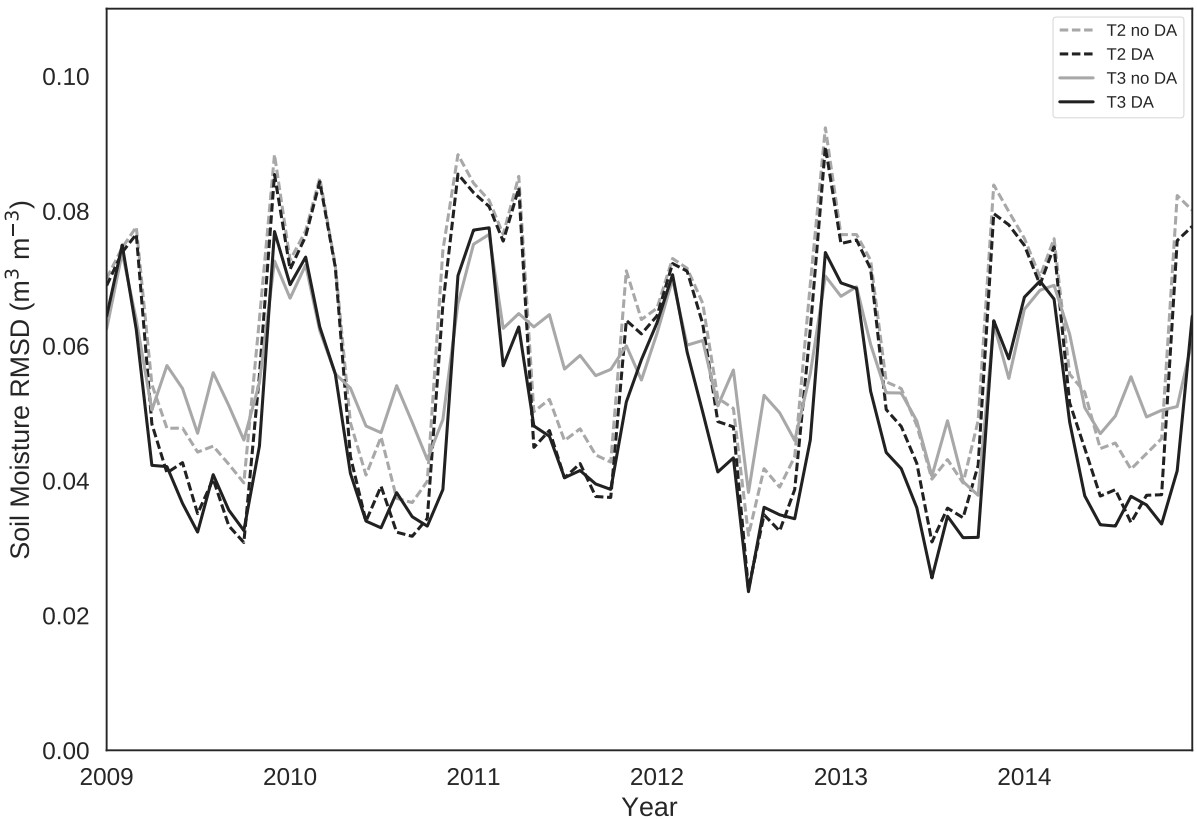

**Figure 7.** Monthly root-mean squared difference (RMSD) of JULES soil moisture estimate compared to ESA CCI for northern Ghana. Light grey dashed line: prior JULES estimate driven with TAMSAT v2.0 precipitation (exp. 1). Dark grey dashed line: prior JULES estimate driven with TAMSAT v3.0 precipitation (exp. 3). Light grey solid line: posterior JULES estimate driven with TAMSAT v2.0 precipitation (exp. 2). Dark grey solid line: posterior JULES estimate driven with TAMSAT v3.0 precipitation (exp. 4).

6% reduction in ubRMSD when calculating statistics over the whole period. In the south of Ghana, we reduce the ubRMSD by 21% from experiment 3 (0.0590 $m^3$ $m^{-3}$) to experiment 4 (0.0467 $m^3$ $m^{-3}$), here improved rainfall data has contributed to 10% of this reduction. We find the highest correlations in the north of Ghana for the whole period (2010 - 2014), this is mainly due to the seasonal cycle being much more pronounced in this region.

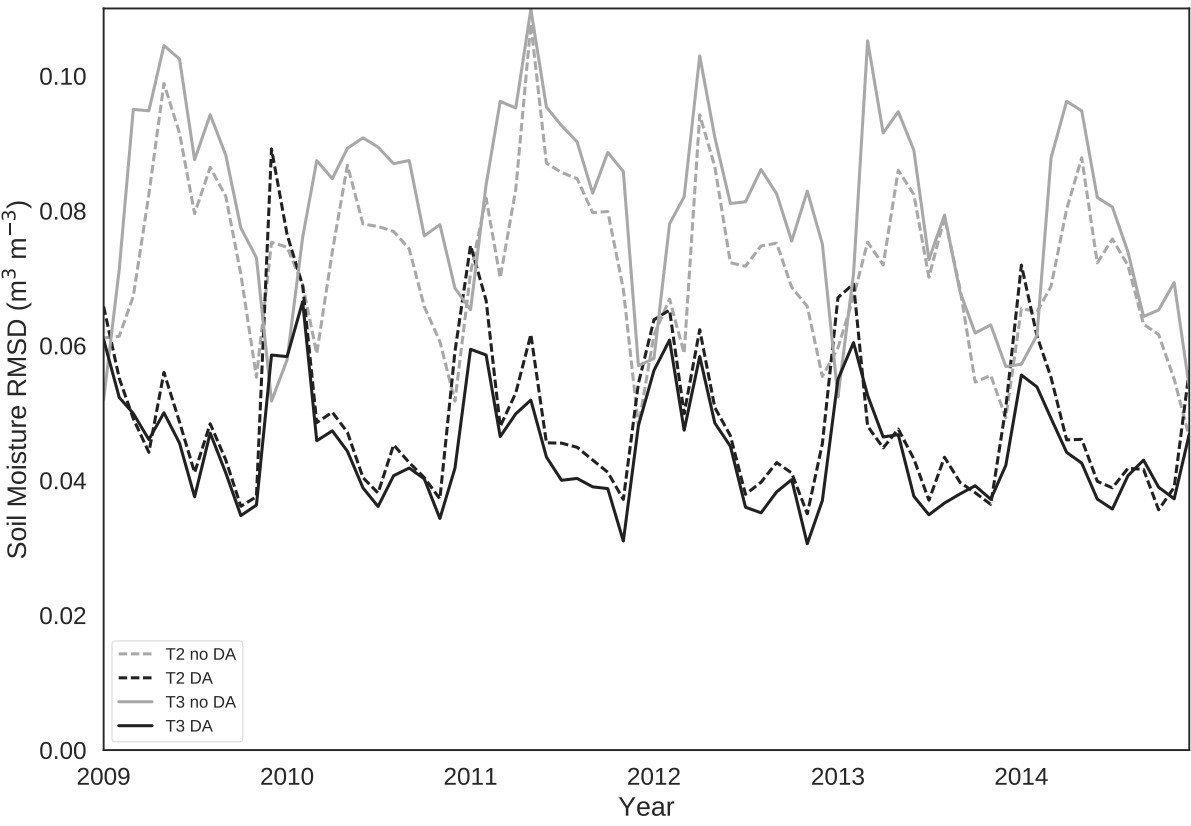

**Figure 8.** As Figure 7, except for southern Ghana.

**Table 1.** RMSE between JULES model soil maps (prior and posterior) and in-situ observations of soil texture from the African Soil Profile Database (Leenaars et al., 2014)

| | Sand RMSE | Silt RMSE | Clay RMSE |
|---|---|---|---|
| North Ghana | | | |
| Prior soil map | 0.43 | 0.25 | 0.30 |
| Posterior soil map | 0.38 | 0.29 | 0.16 |
| South Ghana | | | |
| Prior soil map | 0.35 | 0.27 | 0.16 |
| Posterior soil map | 0.27 | 0.35 | 0.20 |

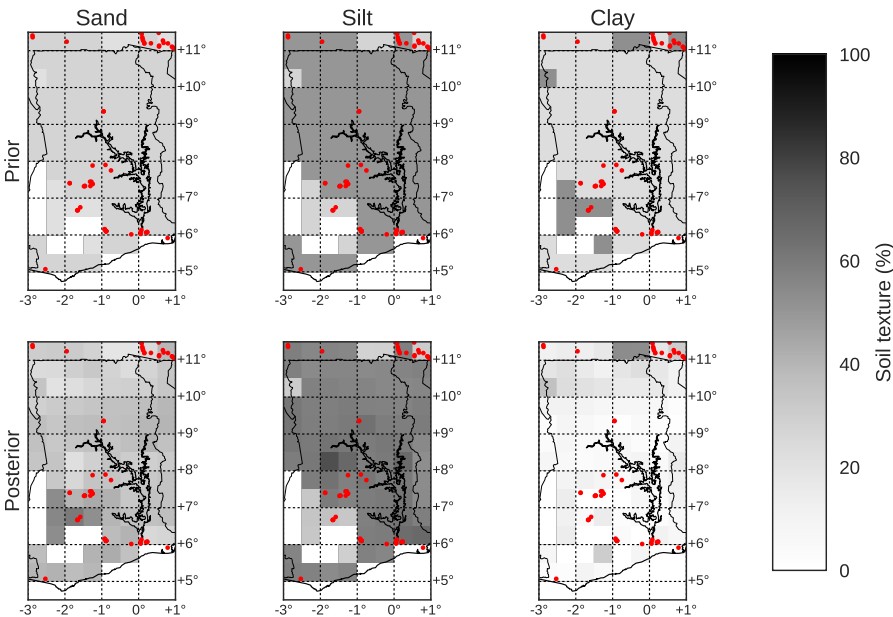

**Figure 9.** Prior and posterior soil maps over Ghana showing percentage of sand, silt and clay. Red dots represent locations where in-situ observations of soil texture are available from The African Soil Profile Database (Leenaars et al., 2014)

**Table 2.** Experiment statistics calculated over the north and south of Ghana in the hindcast period (2010-2014), for the whole period, wetting-up (Mar-May) and drying down (Nov-Jan). The ubRMSD is calculated as $\sqrt{\frac{1}{N}\sum_{i=0}^{N}((\theta_{mod_i} - \bar{\theta}_{mod}) - (\theta_{obs_i} - \bar{\theta}_{obs}))^2}$ with $N$ the number of observations, $\theta_{mod_i}$ the model estimate at time $i$, $\bar{\theta}_{mod}$ the mean model estimate over the time window, $\theta_{obs_i}$ the observation estimate at time $i$ and $\bar{\theta}_{obs}$ the mean observation estimate over the time window. The units of ubRMSD are m$^3$ m$^{-3}$.

| North Ghana | | | | | | | | |
|---|---|---|---|---|---|---|---|---|
| | 1) TAMSAT 2 no DA | | 2) TAMSAT 2 DA | | 3) TAMSAT 3 no DA | | 4) TAMSAT 3 DA | |
| | ubRMSD | Correlation | ubRMSD | Correlation | ubRMSD | Correlation | ubRMSD | Correlation |
| Whole period | 0.0605 | 0.86 | 0.0541 | 0.89 | 0.0622 | 0.86 | 0.0508 | 0.90 |
| Wet | 0.0643 | 0.59 | 0.0592 | 0.64 | 0.0626 | 0.58 | 0.0529 | 0.65 |
| Dry | 0.0396 | 0.77 | 0.0332 | 0.83 | 0.0486 | 0.78 | 0.0365 | 0.84 |
| South Ghana | | | | | | | | |
| | 1) TAMSAT 2 no DA | | 2) TAMSAT 2 DA | | 3) TAMSAT 3 no DA | | 4) TAMSAT 3 DA | |
| | ubRMSD | Correlation | ubRMSD | Correlation | ubRMSD | Correlation | ubRMSD | Correlation |
| Whole period | 0.0651 | 0.77 | 0.0519 | 0.82 | 0.0590 | 0.76 | 0.0467 | 0.82 |
| Wet | 0.0629 | 0.57 | 0.0515 | 0.67 | 0.0571 | 0.55 | 0.0472 | 0.66 |
| Dry | 0.0642 | 0.82 | 0.0492 | 0.85 | 0.0604 | 0.83 | 0.0432 | 0.87 |

## 4 Discussion

For northern Ghana there exists a prominent seasonal cycle for soil moisture, with observations of higher quality than in the south for both TAMSAT rainfall and ESA CCI soil moisture. We find that soil moisture estimates based on TAMSAT v3.0 outperform v2.0, especially during the wetting-up phase of the seasonal cycle, with the effect of the rainfall dataset less marked during the drying down phase. This is to be expected as little or no rain occurs during drying-down so that it is model dynamics that are the dominant factor in the estimation of soil moisture. Therefore, it is the updating of soil parameters via data assimilation and not improved precipitation that has the greatest impact on soil moisture estimates during drying-down. Conversely improved rainfall data has greatest impact for estimating wetting-up and constraining the start of the growing season. This can be seen in Figure 7 where TAMSAT v3.0 without DA outperforms TAMSAT v2.0 with DA at certain times in the season. This is because at these times the data assimilation system is not able to overcome the errors in the precipitation forcing data to improve the estimates further. If there is too little rainfall, there is a point where the DA system cannot make the soil any wetter because we are not changing the model soil moisture state - only the soil texture. Assimilation of CCI soil moisture estimates in the north of Ghana allows us to recover improved estimates to soil texture when judged against in-situ data from the African Soil Profile Database (Leenaars et al., 2014).

For southern Ghana, there exists a much less prominent seasonal cycle than in the north, with poorer quality observations for both TAMSAT rainfall and ESA CCI soil moisture. This is due to large amounts of coastal convective cloud and higher vegetation cover. We find that, after assimilating soil moisture data, runs forced with TAMSAT v3.0 outperform those forced with TAMSAT v2.0. Although we do not have reliable precipitation observations in the south we can still greatly improve our forecast skill for soil moisture through DA. This bodes well for other regions with unreliable precipitation observations (Crow, 2003). In the south we find larger reductions in ubRMSD than in the north after data assimilation. However, we also have less confidence in the CCI soil moisture product to which we are comparing in the south. It is therefore unlikely that we have improved estimates more than in the north in comparison to the truth. This is backed up by the inability of our data assimilation system to recover an improved soil map when compared to in-situ observations in the south.

There is likely an issue of representativity between the satellite derived soil moisture observations and the JULES modelled soil moisture in our DA system. We make the pragmatic assumption that satellite soil moisture is representative of the top 5 cm layer of soil in JULES. However, during intense dry periods the satellite will become more sensitive to greater depths (Ulaby et al., 1982) and hence less representative of the JULES top level soil moisture. This can be seen in Figure 4 where the model fails to capture the satellite observations during the driest periods, with the JULES model predicting a lower soil moisture than the ESA CCI observations, this same phenomenon appears at a number of grid cells during dry periods. We can also see this consistent dry bias in the bottom row of Figure 6b. More work is needed to understand how best to address this issue between satellite and modelled soil moisture. One option could be to create a multi-layer observation operator for land surface models. Previous DA studies have opted to assimilate satellite retrieved brightness temperature and then use a radiative transfer model on top of their chosen land surface model (Moradkhani et al., 2005; Qin et al., 2009; Montzka et al., 2011; Rasmy et al., 2011; Sawada and Koike, 2014; Yang et al., 2016).

Our results highlight the importance of having quality observations of both precipitation and soil moisture. TAMSAT rainfall observations and the ESA CCI soil moisture data are available as daily products but at different spatial resolutions and different observation times. TAMSAT data are produced at 4km spatial resolution by calculating cold cloud duration over a 5 day period of 15 minute thermal infra–red observations. The ESA CCI soil moisture data on the other hand are merged from various passive and active microwave observations and available in various spatial resolutions that are typically in the order of $0.25°$. The core observation that make up the daily product are, in effect, instantaneous but then merged into a harmonised product. The ideal situation would be to have precipitation measurements and soil moisture observations that are representative of the same time periods and spatial domains, but there are no such current missions.

## 5    Conclusions

Previous studies at the grid cell level have shown that calibrating land surface models with satellite observations improves performance when judged against in-situ observations (Moradkhani et al., 2005; Qin et al., 2009; Montzka et al., 2011; Rasmy et al., 2011; Sawada and Koike, 2014; Yang et al., 2016). In this study we calibrated the JULES land surface model at the regional scale (over Ghana) and show that this reduces ubRMSD and correlation when judged against independent observations in a set of hindcast experiments. From the results, it is clear that both improved rainfall estimates and the implementation of data assimilation are required in order to improve modelled estimates and forecasts of soil moisture. We have split our analysis between north and south Ghana due to the hydrological regimes varying considerably between these two regions. In the north of Ghana, where the observations are of highest quality due to lower cloud and vegetation cover, we find that improved precipitation estimates are of greatest importance for accurate representation of the start of season soil moisture. In contrast, the assimilation of relevant soil moisture observations with our land surface model gives the largest benefit for improving estimates during drying-down. This makes physical sense as when no rain is occurring it will be model dynamics that are the dominant factor in the estimation of soil moisture. After data assimilation we are able to improve our estimates of soil texture in the north, judged against in-situ observations. After assimilation of a single year of soil moisture observations (2009) we reduce the ubRMSD of a 5-year model hindcast (2010-2014) by 18% in northern Ghana and 21% in the south, with the improved rainfall product contributing a 6% and 10% reduction in ubRMSD respectively. The higher reduction in ubRMSD in the south is not necessarily indicative of a soil moisture estimate closer to the truth as we also have less faith in the ESA CCI soil moisture product in this region due to higher amounts of convective cloud and vegetation cover. This is supported by the fact that in the south we are unable to recover an improved estimate of soil texture after data assimilation, when judged against in-situ observations. However, in the north we do recover improved soil texture estimates despite the lower reduction in ubRMSD.

*Code availability.*  https://github.com/Ewan82/JULES_DA_Ghana

*Competing interests.* No competing interests are present

*Acknowledgements.* This work was funded by the UK Natural Environment Research Council (NE/P015352/1). This work was also partly funded by the National Centre for Earth Observation. ECB is supported by the Natural Environment Research Council/Global Challenges Research Fund programme ACREW (NE/R000034/1).

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
