# Peer review of "Impact of remotely sensed soil moisture and precipitation on soil moisture prediction in a data assimilation system with the JULES land surface model"

_Hydrology and Earth System Sciences, 2017_

## Referee Comment (RC1) · Anonymous Referee #1 · 30 Dec 2017

This study shows the feasibility of improving soil moisture simulation through parameter calibration based on data assimilation (DA). It further identifies the relative importance of improved rainfall data and DA for soil moisture simulation in wet and dry seasons, i.e., improved rainfall data is more important in wet season but DA is more important in dry season. In satellite era, I anticipate assimilating globally available data to estimate model parameter values should be a new and promising way to improve land hydrological simulations and therefore this kind of studies should be encouraged. Although this study is not the first one in terms of this topic, it presents a case to demonstrate the effectiveness of this way. The paper is well organized and written clearly. I would like to see its publication after minor revisions below.

(1) The framework of the DA used in this study is very similar to Yang et al. (2007; 2009). This DA framework was used by Rasmy et al. (2011) and Sawada et al. (2014). The strategy to select optimized parameters (P4L31-32, P5L1-2) is also similar to Yang et al. (2007).

(2) Did the authors optimize soil porosity? As demonstrated in Yang et al. (2016), soil moisture estimation is most sensitive to this parameter, and in turn, this parameter is the most possible one to be estimated reasonably. A sensitivity analysis to include soil porosity as an optimized parameter can help our understanding.

(3) P6L28-29: "a finding consistent with the comparisons of precipitation between v3.0 and v2.0 presented by Maidment et al. (2017)". This is not clear if not referring to this paper. It is expected to give a little explanation to the finding in Maidment et al. (2017).

(4) P8L3-4: "it is possible that for other grid cells we are overfitting to the data.", I don't think so, because there are only two parameters optimized, as sand%+silt%+ clay%=100%. The pedo-transfer functions are empirical, and the optimized parameter values are apt to satisfying better soil moisture estimate instead of soil texture estimate.

(5) P8L10-13, I don't agree the discussion. The biases are so small that are within uncertainties of CCI soil moisture data.

(6) P12L23-25. "representativity between the satellite derived soil moisture observations and the JULES modelled soil moisture in our DA system". The CCI soil moisture is believed to be representative of a much shallower layer than 10 cm.

Ref.
Yang et al., 2007. Auto-calibration system developed to assimilate AMSR-E data into a land surface model for estimating soil moisture and the surface energy budget. J. Meteorol. Soc. Jpn. 85A, 229–242.
Yang et al., 2009. Validation of a dual-pass microwave land data assimilation system for estimating surface soil moisture in semiarid regions. J. Hydrometeorol. 10, 780–793.

---

## Short Comment (SC1) · 3 Jan 2018

Dear Authors

Please can you confirm whether the ESA CCI combined soil moisture product used in this study has the same Cumulative Distribution Function (CDF) as the GLDAS-NOAH model? If so, will the ESA-CCI product have the same biases as the GLDAS-NOAH model? Therefore, does it make sense to use bias and Root Mean Square Error (RMSE) as your verification metrics? Your Table 1 does suggest that the parameter optimisation is helping to reduce the random errors in your experiment. Therefore, would it be much better to use verification metrics such as temporal correlation and

unbiased Root-Mean-Squared-Difference (ubRMSD)?

The reasons why the ESA CCI combined product is matched to have the same CDF as the GLDAS-NOAH model are given in Dorigo et al 2017 and references therein.

It would be highly desirable to repeat your parameter optimisation experiment over data rich areas such as portions of the continental United States and Australia. This would allow the use of ground based soil moisture observation for verification. You may wish to consult with Dharssi and Vinodkumar (2017) for availability of Australian ground based soil moisture observations and validation. Also the STATSGO (Miller and White, 1998) soil map could be used to evaluate your optimised sand/silt/clay soil fractions for the US.

For figures 2 and 3, please can you clarify whether the "observations" are the ESA-CCI product. If so, please label accordingly.

Dorigo, W., Wagner, W., Albergel, C., Albrecht, F., Balsamo, G., Brocca, L., ... & Haas, E. (2017). ESA CCI Soil Moisture for improved Earth system understanding: state-of-the art and future directions. Remote Sensing of Environment, 203, 185-215.

Miller, D. A., & White, R. A. (1998). A conterminous United States multilayer soil characteristics dataset for regional climate and hydrology modeling. Earth interactions, 2(2), 1-26.

Dharssi, I and Vinodkumar. (2017). JASMIN: A prototype high resolution soil moisture analysis system for Australia. Bureau Research Report No. 026. Melbourne, Australia. http://www.bom.gov.au/research/publications/researchreports/BRR-026.pdf

Sincerely

Imtiaz Dharssi

---

## Referee Comment (RC2) · G.V. Ayzel (Referee) · 12 Jan 2018

General comments: A lot of remote sensing products of hydrometeorological variables are available today for the scientific community. There are numerous studies check remote sensing data consistency with ground-based observations, evaluate their efficiency as forcing to LSM or more specific models, investigate a possibility for using this data as an additional source for making our models more realistic. At first glance (looking at title and abstract), the reviewed article tries to show a benefit of including remote sensing soil moisture data to enhancing JULES model realism in modeling various water budget components, but during the reading, my impression was changing

from positive to indifferent. The main reason for this transition is a discrepancy between the title (which sounds novel) and the content. In my humble opinion, a title such as "Top-layer soil moisture prediction based on JULES model and remote sensing data using 4D-Var inspired cost function for soil content parameters calibration" would fit this content best. The only (and minor) difference of classical approaches for water balance modeling using LSM models (e.g. in Zulkafli et al., 2013; Le Vine et al., 2016) here is using the only soil moisture as a target variable. There is no real procedure for water budget constraining (e.g. as in Kelleher et al., 2017). Obviously, there is no discussion about water budget components except soil moisture. In my opinion, the presented article needs a major revision to be published in HESS.

Specific comments: P1L1-2, P1L10, P2L29: There are direct links of presented research to the problem of drought assessment for the agricultural sector, but there is no place in the article for any drought index calculation or introducing a transition from better soil moisture estimates to better decision making in crops cultivation. Authors need to eliminate this part from an abstract or complement Results, Discussion, and Conclusions part of respective information. P1L5-6: Rough estimations of data sources assimilation contribution based on spatially averaged data – need to be reconsidered for the North and the South part separately. P10: There is no reason to write about soil moisture forecasting, because of an experimental design setting is build around non-operational forcing (WFDEI). P2L13-14: What is the novelty of presented research in comparison with Bolten et al., 2010? P3L4-5: Model was driven only by TAMSAT reanalysis products, there was no observational (measured on a station) forcing included. P4L31: Strictly speaking authors updated only two parameters of soil texture, not three. P6L9: "In Figure 2 we show the results of a data assimilation and forecast for a single grid cell in the north of Ghana", but Figure 2 caption is "Soil moisture data assimilation results for a north Ghana grid box". Please, unify this parts (the same in Figure 3 and P6L17). P6L13-14: For me, it is obvious that an amplitude of modeling results is decreasing from 2010 to 2014 (maybe insignificant). Is there any pattern in precipitation of other forcing variables corresponding with this behavior? P6L16-17: It

would be great to show a skill of JULES over daily averages of ESA CCI data for 2009 which can be used as "climatological" soil moisture scenario for long-term predictions and forecasting. The same for Figure 3 (results for the south part). P6L20-21: "Although we do improve the fit to the observations after data assimilation in Figure 3..." – not so clear statement without a reference to quantitative metric. P6L30-31: It would be great to see an additional figure of spatial differences between TAMSAT versions for clearer understanding drivers of soil moisture patterns. P6L32: The choice of yearly root-mean-square error (despite considering wet/dry seasons) significantly reduces results understanding in seasonal dynamics. Transfer to monthly (or even weekly) can help both authors and readers to understand dynamic patterns. P7L12-15: Is there another global database of soil properties for comparison with HWSD and JULES-DA results? P8L5-12: Calculation of summary statistics over the whole of Ghana can corrupt obtained results and conclusions – it is a clear pattern (shown in Figure 4) for both different amplitude and direction of bias between southern and northern parts of Ghana. It is better (and will be consistent with results presented in Figures 2, 3, 5) to calculate summary statistics separately for southern and northern parts and presented results as a figure. Maybe it will cause different results and further discussion. P12L5-8: It would be great to describe in details why "updating of soil parameters" (= updating all of the parametrization of water cycle processes in JULES) sometimes works worse than precipitation forcing updating. P12L12-14: True for any calibration approach. But it is not possible to improve anything without a forcing data. P12L15-22: This part is too technical and looks out of context for the Discussion section. P13L9-19: Conclusions are too technical and do not provide any information about the progress in "the understanding of hydrological systems, their role in providing water for ecosystems and society, and the role of the water cycle in the functioning of the Earth system." (quote from the HESS aims and scope).

Typos: P12L11: "after after".

References: Kelleher, C., McGlynn, B., and Wagener, T.: Characterizing and reducing equifinality by constraining a distributed catchment model with regional signatures, local observations, and process understanding, Hydrol. Earth Syst. Sci., 21, 3325-3352, https://doi.org/10.5194/hess-21-3325-2017, 2017. Le Vine, N., Butler, A., McIntyre, N., and Jackson, C.: Diagnosing hydrological limitations of a land surface model: application of JULES to a deep-groundwater chalk basin, Hydrol. Earth Syst. Sci., 20, 143-159, https://doi.org/10.5194/hess-20-143-2016, 2016. Zulkafli, Z., Buytaert, W., Onof, C., Lavado, W., and Guyot, J. L.: A critical assessment of the JULES land surface model hydrology for humid tropical environments, Hydrol. Earth Syst. Sci., 17, 1113-1132, https://doi.org/10.5194/hess-17-1113-2013, 2013. critical assessment of the JULES land surface model hydrology for humid tropical environments, Hydrol. Earth Syst. Sci., 17, 1113-1132, https://doi.org/10.5194/hess-17-1113-2013, 2013.

---

## Referee Comment (RC3) · CM Massari (Referee) · 30 Jan 2018

Review of the paper:

**Using satellite observations of precipitation and soil moisture to constrain the water budget of a land surface model**

by Ewan Pinnington et al.

**General comments**

The paper describes a data assimilation exercise in which CCI satellite soil moisture observations derived from the active+passive product are ingested into the JULES land surface model. The 4D variational data assimilation approach is used for the adjustment of model parameters via the correction of the soil texture (i.e., the relative percentage of clay, sand and silt) through pedo-transfer functions. The authors show that after the ingestion the model soil moisture estimates are closer to CCI observations not only during the assimilation time period but also in the hindcast period. In addition, TAMSAT precipitation is used for driving the model suggesting that a better precipitation product is able to improve the soil moisture wetting up better than the assimilation of soil moisture does.

**Evaluation**

The paper is well written and clear and well fits the scope of the journal. I have four main MAJOR issues that the authors should clarify prior the paper can be considered adequate for a publication in HESS. I also have additional comments that should be addressed.

1) My main concern about this DA exercise relies upon the validation procedure. A fundamental assumption made by the authors (but never cited in the text) is that the CCI soil moisture product, being an observation, can be considered close to the truth. This is not explicitly said in the text but the numerous "RMSE reduction", "bias reduction" statements make think that we are reducing an error with something that is supposed to be very accurate and basically a close representation of the true soil moisture at the ground. However, satellite soil moisture products are far to be close to the truth, especially in areas with dense vegetation. Based on that, the exercise appears to me more a way to adjust the JULES soil moisture estimates to the CCI observations rather than an effective and real improvement of the model estimates. To this end, I suggest to:

- Demonstrate with independent observations (i.e., for instance derived from in situ stations) that the CCI is a relatively good product in Ghana and that the soil moisture estimates after assimilation are able to improve the JULES soil moisture estimates (and possibly the CCI itself which is the main task of any assimilation exercise). If independent observations are not available the authors can cross-validated the different soil moisture products by using them within an application.

- Change notation when compare JULES estimates with CCI as RMSE and BIAS refer to a product, i.e., the CCI, that is already uncertain itself and has its own bias and error with an unknown truth. I suggest to use root mean square differences (RMSD) and mean relative error.

2) A second issue is related to the performance scores used by the authors. That is RMSE and BIAS. From the figures it clearly appears that the main contribution of the DA is the correction of the bias. Therefore, the reduction in RMSE which contains information about both the correlation and the bias is mainly driven by the bias adjustment. That is, assuming CCI a good representation of the soil moisture at the ground, it is not clear if the DA is able to reduce the random error or only change the bias. If this is the case this could be simply achieved by a simple rescaling of the model estimates to the CCI observations. Therefore, I suggest the authors to use self-consistent scores like correlation, $R^2$ or fractional root mean squared differences (fRMSD, Draper et al. 2013).

Draper, C., Reichle, R., de Jeu, R., Naeimi, V., Parinussa, R., & Wagner, W. (2013). Estimating root mean square errors in remotely sensed soil moisture over continental scale domains. *Remote Sensing of Environment*, *137*, 288-298.

3) A third issue, also highlighted by the authors but only at the end of the manuscript is the depth mismatch between ingested observations and model estimates. I think that the assumption of CCI soil moisture observations being representative of the first 10 cm is unrealistic. I suggest reformulating this assumption or providing more robust evidences for demonstrating it. In this respect, I have a suggestion. If the authors want to ingest CCI observations and solve the problem of the depth mismatch between CCI and JULES a simple and effective solution could be the use of the exponential filter (Albergel et al. 2008). Many studies of sequential DA into hydrological model use this solution (see Massari et al. 2015, Alvarez and Garreton 2016).
In addition there is not mention about the effect of the ingested observations in the deeper layers. This should at least mentioned and discussed.

*Albergel, C., Rüdiger, C., Pellarin, T., Calvet, J. C., Fritz, N., Froissard, F., ... & Martin, E. (2008). From near-surface to root-zone soil moisture using an exponential filter: an assessment of the method based on in-situ observations and model simulations. Hydrology and Earth System Sciences Discussions, 12, 1323-1337.*

*Massari, C., Brocca, L., Tarpanelli, A., & Moramarco, T. (2015). Data assimilation of satellite soil moisture into rainfall-runoff modelling: A complex recipe? Remote Sensing, 7(9), 11403-11433.*

*Alvarez-Garreton, C., Ryu, D., Western, A. W., Crow, W. T., Su, C. H., & Robertson, D. R. (2016). Dual assimilation of satellite soil moisture to improve stream flow prediction in data-scarce catchments. Water Resources Research, 52(7), 5357-5375.*

4) Finally yet importantly, the changing of the soil texture after DA should somehow reflect the real ground texture more than the Harmonized database can do. This has to be demonstrated and

can constitute an additional proof for the DA exercise to be able to improve the model representation of the reality. Otherwise, DA becomes a simple fitting of the CCI observations.

I have also other additional comments that I will list below in order of appearance in the manuscript.

MODERATE. Section 2.2 Define here which are the differences between TAMSAT 2.0 and 3.0

MODERATE. Equation 1. Define $N$. Also $x_i$ should not be $x_o$? What does $x_i$ represent?

MAJOR. Section 2.4. Describe here or later how the matrices $B$ and $R$ are estimated. This is totally missing in the text and is one of the most important topic in data assimilation.

Pag 12. Line 15. To retrieve hydraulic parameters… There is not proof in the paper that the DA scheme used is able to retrieve hydraulic parameters. Please clarify this aspect or remove.

---

## Referee Comment (RC4) · W. Dorigo (Referee) · 4 Feb 2018

This study is an interesting contribution of how satellite soil moisture an precipitation products can be used to improve land surface model estimates in data-poor regions. However, I have a few major concerns regarding the use of the ESA CCI Soil Moisture data. These have been adequately expressed by the reviews of Imraz Dharssi and Christian Massari, so I will not repeat them here. All these issues stem from the fact that absolute soil moisture values of the ESA CCI Soil moisture COMBINED product are strongly influenced by the climatology of GLDAS1-Noah soil moisture. A starting point for improving your study could therefore be to first look at metrics reflecting the

similarity in the temporal domain (e.g., correlation).

In your discussion about the differences in product skill between the South and the North (Figures 2 and 3) you mention that these products have lower skill in the south. How then can you explain the larger improvement and lower overall errors shown for the South in Figure 5?

Could you please indicate the version of ESA CCI Soil Moisture you have used?

---

## Author Comment (AC1) · 28 Mar 2018

We thank the reviewer for their comments which have helped strengthen the manuscript. Please find attached a zip file containing a pdf of our responses and a pdf of an updated manuscript with proposed changes highlighted in blue.

The response document is structured with reviewers comments in black text and author responses in blue text.

Kind Regards, Ewan Pinnington

[Figure]

Please also note the supplement to this comment:
https://www.hydrol-earth-syst-sci-discuss.net/hess-2017-705/hess-2017-705-AC1-supplement.zip

———————————————————

---

## Author Comment (AC2) · 28 Mar 2018

We thank the author for their short comment which has helped strengthen the manuscript. Please find attached a zip file containing a pdf of our responses and a pdf of an updated manuscript with proposed changes highlighted in blue.

The response document is structured with reviewers comments in black text and author responses in blue text.

Kind Regards, Ewan Pinnington

[Figure]

Please also note the supplement to this comment:
https://www.hydrol-earth-syst-sci-discuss.net/hess-2017-705/hess-2017-705-AC2-
supplement.zip

---

## Author Response (AR1)

**HESS review responses**

**Author response to R1**

We thank the reviewer for their kind comments which have helped to strengthen this manuscript. Below are our responses:

1) The reviewer noted that the DA framework in this study is very similar to previous studies.

We agree that the method is similar to the studies noted, we have updated the text to clarify this. P7L7:

"This is a similar framework to that introduced in Yang et al. (2007, 2009) for data assimilation with the Simplified Biosphere model 2 (SiB2) (Sellers et al., 1996), …"

2) The reviewer asked if we had optimized soil porosity.

Soil porosity is updated implicitly via the pedo-transfer functions we use for the JULES model. We have updated the text to make this clear. P7L8:

"…with the exception that the soil porosity parameter of JULES is updated implicitly within the pedo-transfer functions rather than explicitly included in the optimisation."

3) The reviewer asked for more explanation of the reference to *Maidment et al. 2017* P6L28-29.

We have updated the texted with more explanation. P8L30:

"This finding is consistent with the comparisons of precipitation between v3.0 and v2.0 presented by Maidment et al. (2017), where TAMSATv3.0 was shown to reduce a dry bias present in TAMSATv2.0 when compared to ground station data."

4) The reviewer disagreed that we were overfitting to the data at some grid cells P8L3-4.

We agree that "overfitting" is not the right terminology here. We believe that missing processes within the model and deficiencies of our pedo-transfer functions may cause us to retrieve unrealistic soil texture values for some grid cells. We have updated the text to remove the previous explanation. We have also included a comparison to in-situ soil texture observations from the African Soil Profile Database. P11L10:

"The inability of our data assimilation to improve soil texture estimates at certain points is most likely due to issues of spatial representativity between the modelled soil map and the in-situ data. It is also possibly impacted by errors in our pedo-transfer functions, which may perform better if specifically calibrated for Ghanaian soils (Patil and Singh, 2016)."

5) The reviewer didn't agree with our discussion on P8L10-13.

We agree with the reviewer that discussion about the biases was not meaningful and have removed this discussion from the text.

6) The reviewer commented that the CCI soil moisture observations are believed to be representative of a depth of less than 10cm P12L23-25.

Reviewer 3 also thought that 10cm was too deep a layer for comparison with the CCI soil moisture. We have therefore updated the JULES model to run with a top layer of 5cm depth and re-run our experiments. P3L15:

"JULES is typically run with 4 soil layers, with the top layer being 10 cm deep. In this paper we have updated JULES to run with a top layer of 5 cm to be more representative of the ESA CCI soil moisture observations. Another option to deal with the issue of representativity would be an exponential filter (Albergel et al., 2008) which has been used in sequential data assimilation studies previously (Massari et al., 2015; Alvarez-Garreton et al., 2016)."

**Author response to R2**

We thank the reviewer for their comments which have certainly helped to strengthen the manuscript. Please find our responses below:

1) The reviewer suggested that we change the title of the paper as the previous title was not well aligned with the paper contents.

We agree with the reviewer that a more appropriate title is necessary and have updated the this accordingly:

"Impact of remotely sensed soil moisture and precipitation on soil moisture prediction for a data assimilation system with the JULES land surface model"

We feel that this more accurately brings out the novel aspects of the paper, i.e. and exploration of the useful information two different EO data sources – one driving the model (precipitation) and the other being used as a target variable (soil moisture).

2) The reviewer asked us to either remove discussion of agricultural drought from the abstract or complement the Results, Discussion and Conclusion with how the work here would be useful for better decision making in crop cultivation.

We agree that agricultural drought is not discussed enough throughout the paper to warrant its inclusion in the abstract. We have removed the relevant text.

3) The reviewer asked that we include the reduction in root-mean squared differences for both the North and South within the abstract P1L5-6.

We have included the statistics for the two regions instead of the whole of Ghana. We have also updated the performance statistic used at the request of other reviewers to unbiased root-mean squared error. P1L5:

"The use of an improved remotely-sensed rainfall dataset contributes to 6% of this reduction in error in northern Ghana and 10% in southern Ghana."

3) The reviewer asked us to remove mention of forecasting from the abstract P1L10

We have removed this from the abstract

4) The reviewer asked us to comment on the novelty of this research, compared to *Bolton et al. 2010* P2L13-14.

Bolten et al. 2010 conduct state estimation using a sequential data assimilation technique (Ensemble Kalman Filter), meaning that whenever an observation is available the model state (in this case soil moisture) will be updated, but if no data is available the model estimate to soil moisture will not be updated. In this paper we are conducting parameter estimation using a variational technique. Once we have ingested data over some time window the model parameters governing soil moisture will be updated and so will future and past estimates to soil moisture. We have clarified this in the text. P2L16:

"It has been shown by Bolten et al. (2010) that assimilation of remotely sensed surface soil moisture can significantly improve the prediction of root-zone soil moisture and drought modelling, where a sequential DA technique is used for soil moisture state estimation."

5) The reviewer asked us to remove the term "observations" from P3L4-5.

We have removed this.

6) The author asked us to highlight that we update only two parameters of soil texture, with the third being updated implicitly P4L31.

We have amended the text to show this. P7L5:

"we updated the percentage of sand and silt in the soil at each minimisation step (with clay being updated implicitly)"

7) The reviewer asked us to unify our usage of terminology "grid cell" and grid box" P6L9 and P6L17 with Figure caption 2.

We have updated the text accordingly.

8) The reviewer commented that the amplitude of soil moisture is decreasing for the model (2010-2014) and wondered if the precipitation data showed the same pattern P6L13-14.

We agree there is a noticeable trend in the amplitude of soil moisture and this is also seen in the precipitation data for both TAMSAT products. We have added a comment on this in the paper. P8L15:

"In Figure 4 we can also see the amplitude of this seasonal cycle slightly decreasing, this is a pattern also seen in both TAMSAT products which exhibit a drying over the period 2010-2014 for this grid cell"

9) The reviewer suggested it would be good to show a skill of JULES over daily averages of ESA CCI data for 2009 for Figure 2 and 3.

We have included root-mean squared differences between JULES and CCI data in the text. In relation to Figure 2, P8L13:

"The model skill for predicting this seasonal cycle is markedly improved after data assimilation, with a root-mean squared difference (RMSD) of 0.035 after data assimilation compared to a RMSD of 0.094 before for 2009"

In relation to Figure 3, P8L20:

"(RMSD of 0.059 after data assimilation compared to RMSD of 0.102 before for 2009)"

10) P6L20-21: "Although we do improve the fit to the observations after data assimilation in Figure 3..." – Reviewer asked us to add a quantitative metric.

We have included RMSD reduction as a quantitative metric, please see previous comment (no. 9).

11) The reviewer asked us to add a plot of the difference in precipitation between TAMSAT versions P6L30-31.

We have added the suggested figure (See new Figure 1) and also included a figure of cumulative rainfall over the period of the experiments (2009 – 2014) to show TAMSAT v2.0 is the drier product (see new Figure 2).

We have added text to section 2.2 discussing these figures, P4L15:

"In Figure 1 we show yearly cumulative rainfall averaged over 2009 - 2014 for TAMSAT v2.0 and v3.0, we can see the different spatial distributions of rain with v3.0 being wetter in the south and v2.0 wetter in the east. To show the difference in the amount of rainfall for the two products we also show cumulative rainfall for the period 2009 - 2014 averaged spatially over Ghana in Figure 2, this shows TAMSAT v2.0 to be the drier of the two products overall, as expected."

12) The reviewer asked us to update Figure 5 to have monthly or weekly RMSEs (rather than yearly) P6L32

We have updated this figure accordingly to display monthly rather than yearly averages and now display two new Figures (see Figure 7 and 8 in attached manuscript of proposed changes), as the reviewer has suggested this helps to understand dynamic patterns better. P8L35:

"Figure 7 and 8 show experiment monthly root-mean-square differences (RMSD) for north and south Ghana respectively. For Figure 7 this shows that the most accurate model run overall is experiment 4 (TAMSAT v3.0 with DA). We see in the majority of years that towards the start of the season as soils are wetting up it is experiment 3 and 4 (TAMSAT v3.0 no DA and with DA respectively) that have the lowest RMSD, suggesting that it is precipitation, as

opposed to the assimilation of soil moisture, that is most important for improving soil moisture estimates during this period. This relationship changes towards the end of the rainy season with experiment 2 and 4 being the most accurate (TAMSAT v2.0 with DA and TAMSAT v3.0 with DA respectively) suggesting that assimilation of soil moisture estimates is most important in this period…"

13) The reviewer asked if there was another database of soil properties for comparison with HWSD and JULES-DA results P7L12-15.

We have used in-situ observations from The African Soil Profile Database compiled by ISRIC to compare with our results. We find an improvement in soil texture in the north of Ghana compared to the in-situ measurements. This is not the case for the south, where we slightly degrade our estimate to soil texture. Relevant text has been added to P10L13:

"Comparing estimates of soil texture derived from CCI soil moisture to in-situ observations is inevitably problematic due to spatial scales of representativity. However, independent sources of verification are difficult to find over Ghana. We therefore compare or soil maps to in-situ observations from The African Soil Profile Database (Leenaars et al., 2014). This database is compiled by the International Soil Reference and Information Centre (ISRIC) with the quality of the data being rated from 1 (highest quality) to 4 (lowest quality), here we compare only to observations with a quality flag of 1 or 2. In table 1 we show the root-mean-squared error (RMSE) for our soil maps when compared to 21 in-situ observations of soil texture in the north of Ghana and 36 in-situ observations in the south (locations shown as red dots in Figure 9). For the north of Ghana where we have most confidence in our results we find a reduction in RMSE for both sand and clay (almost halving the RMSE in clay %). However, we do increase our RMSE for the silt %. In the south of Ghana, we do not manage to recover a better estimate to soil texture after data assimilation, with an increase in RMSE for silt and clay but a decrease in RMSE for sand. The inability of our data assimilation to improve soil texture estimates at certain points is most likely due to issues of spatial representativity between the modelled soil map and the in-situ data. It is also possibly impacted by errors in our pedo-transfer functions, which may perform better if specifically calibrated for Ghanaian soils (Patil and Singh, 2016)."

14) The reviewer asked us to calculate summary statistics for the north and south of Ghana rather than the country as a whole P8L5-12

We have updated the summary statistics in Table 2 accordingly. At the request of other reviewers, we have also changed the statistics used in this table from RMSE and bias to unbiased RMSD and correlation.

15) P12L5- 8: The reviewer asked us to describe in details why "updating of soil parameters" (= updating all of the parametrization of water cycle processes in JULES) sometimes works worse than precipitation forcing updating.

We have added more explanation here. P15L10:

"This can be seen in Figure 7 where TAMSAT v3.0 without DA outperforms TAMSAT v2.0 with DA at certain points in the season. This is because at these points the data assimilation system is not able to overcome the errors in the precipitation forcing data to improve the estimates further. If there is too little rainfall, there is a point where the DA system cannot make the soil any wetter (because we are not changing the model soil moisture state - only the soil texture)."

16) The reviewer commented that it is not possible to improve predictions with no access to forcing data P12L12-14.

We have removed this text.

17) The reviewer commented that discussion of the data assimilation system was too technical for the discussion P12L15-22.

We have moved this text to the methods section 2.4

18) The reviewer commented that the conclusions were too technical and miss the HESS aims and scope P13L9-19.

We have updated conclusions to try and address this point

Typos:

P12L11: "after after".

amended.

**Author response to R3**

We thank the reviewer for their comments which have helped to strengthen our manuscript. Please find the responses below:

1) The reviewer suggested that we change the notation from RMSE and BIAS to RMSD and mean relative error as the prior notation makes the assumption that the ESA CCI estimates are close to the truth. The reviewer also commented that we should make an effort to include some independent observations.

We agree that the notation suggested by the reviewer is more appropriate and we have updated the manuscript accordingly. Another reviewer has also raised the concern over the quality of the observations with which we are comparing, especially as they are CDF matched to another land surface model. We have taken their advice and also included temporal correlation and the unbiased RMSD (ubRMSD) as metrics to show that the model still improves after removing any competing bias. We have added text commenting on the issues for the ESA CCI combined soil moisture product. We have also included independent in-situ observations of soil texture from the African Soil Profiles Database to judge our models soil maps and added text discussing the performance of CCI soil moisture over West Africa in section 2.3, P4L9:

"Dorigo et al. (2015) also show that the ESA CCI product performs well over Western Africa when judged against in-situ soil moisture observations from the AMMA network (Cappelaere et al., 2009) with stations in Benin, Mali and Niger. When judged against the AMMA network CCI soil moisture was shown to have a high correlation (~ 0.7) and one of the lowest unbiased root-mean squared differences (~ 0.04) of the 28 worldwide networks used in the study. This bodes well for our comparison over Ghana, which has a similar climate regime in the north to the sites in the AMMA network."

2) The reviewer suggested we included different performance scores as from the two included it appeared that the DA mainly acted to reduce the BIAS (mean relative error).

We agree that an additional performance score would be beneficial. As another comment from the author of a short comment on the paper has also suggested. We have included their suggestion of temporal correlation and ubRMSD to show that the DA technique also reduces random errors within the model, rather than just acting to rescale the model predictions. P11L14:

"Satellite soil moisture products can be subject to larger errors and biases associated with data processing. This is particularly true for the CCI level 3 combined active and passive product used in this paper, as in order to merge information from 11 different sensors data is CDF matched to the GLDAS-Noah v1 model (Rodell et al., 2004). Therefore, any bias within the GLDAS-Noah model will be included in the level 3 soil moisture product used here. To

make sure we are not just correcting the bias of the JULES model to that of GLDAS-Noah we include summary statistics of unbiased root-mean squared difference (ubRMSD) and temporal correlation in table 2. In every case we find that after data assimilation we improve both ubRMSD and correlation and in the majority of cases find the best results for experiment 4 (TAMSAT v3.0 with DA). For the north of Ghana, we reduce the ubRMSD by 18% from experiment 3 (0.0622 m3 m−3) to experiment 4 (0.0508 m3 m−3). From experiment 2 to 4 we can see that, after data assimilation, using TAMSAT v3.0 rainfall over v2.0 has contributed to a 6% reduction in ubRMSD when calculating statistics over the whole period. In the south of Ghana we reduce the ubRMSD by 21% from experiment 3 (0.0590 m3 m−3) to experiment 4 (0.0467 m3 m−3), here improved rainfall data has contributed to 10% of this reduction."

3) The reviewer commented that the assumption made that CCI soil moisture observations are representative of the JULES top 10cm soil layer is unrealistic. They suggested we reformulate this assumption or use an exponential filter to address this.
We agree that the assumption made originally is perhaps not realistic. We have therefore updated the JULES model to run with a 5cm soil top layer and have re-run our experiments. We have updated the text in section 2.1 to reflect this and have also included the references mentioned by the reviewer commenting that this is another option. We find similar results as when running with a 10cm soil depth except a larger dry mean relative error in the north after data assimilation (see Figure 6). This is understandable as a shallower soil layer will dry more quickly. P3L16:
"In this paper we have updated JULES to run with a top layer of 5 cm to be more representative of the ESA CCI soil moisture observations. Another option to deal with the issue of representativity would be an exponential filter (Albergel et al., 2008) which has been used in sequential data assimilation studies previously (Massari et al., 2015; Alvarez-Garreton et al., 2016)."

4) The reviewer asks us to demonstrate the soil texture after DA more closely reflects reality than the HWSD.
We have included comparisons of our retrieved soil maps to in-situ observations of soil texture from the African soil profile database. We find that we can improve soil texture estimates in the north but are unable to do so in the south. We have added discussion about this in the text. P10L13:
"Comparing estimates of soil texture derived from CCI soil moisture to in-situ observations is inevitably problematic due to spatial scales of representativity. However, independent sources of verification are difficult to find over Ghana. We therefore compare or soil maps to in-situ observations from The African Soil Profile Database (Leenaars et al., 2014). This database is compiled by the International Soil Reference and Information Centre (ISRIC) with the quality of the data being rated from 1 (highest quality) to 4 (lowest quality), here we compare only to observations with a quality flag of 1 or 2. In table 1 we show the root-mean-squared error (RMSE) for our soil maps when compared to 21 in-situ observations of soil texture in the north of Ghana and 36 in-situ observations in the south (locations shown as red dots in Figure 9). For the north of Ghana where we have most confidence in our results we find a reduction in RMSE for both sand and clay (almost halving the RMSE in clay %). However, we do increase our RMSE for the silt %. In the south of Ghana, we do not manage to recover a better estimate to soil texture after data assimilation, with an increase

in RMSE for silt and clay but a decrease in RMSE for sand. The inability of our data assimilation to improve soil texture estimates at certain points is most likely due to issues of spatial representativity between the modelled soil map and the in-situ data. It is also possibly impacted by errors in our pedo-transfer functions, which may perform better if specifically calibrated for Ghanaian soils (Patil and Singh, 2016)."

5) Section 2.2 Define here which are the differences between TAMSAT 2.0 and 3.0

We have added text explaining the difference between the 2 products and added two figures showing how the products differ over Ghana. P4L10:

"TAMSAT v3.0 differs from TAMSAT v2.0 in that it uses an updated calibration against in-situ data that is more representative of local scales. […] For more information on the differences between the two TAMSAT products see Maidment et al. (2017)."

See new Figures 1 and 2 in attached manuscript of proposed changes.

6) Equation 1. Define N. Also xi should not be xo?

The reviewer is correct xi should be x0, N is the number of observations within the chosen time window. We have updated the text to clarify this.

7) Section 2.4. The reviewer asked us to describe how the matrices B and R are estimated.

We have described how B and R are specified in this section. P6L13:

"As we do not have a good estimate to the error in the prior estimates of model parameters we chose a conservative 5% standard deviation for the prior error covariance matrix B. This ensures we do not retrieve unrealistic estimates of soil texture after data assimilation. For the observational error covariance matrix R we have a diagonal matrix with variances estimated from the standard deviations included in the ESA CCI soil moisture product."

8) P12L15 The reviewer asked us to remove the phrase "retrieve hydraulic parameters".

Removed

**Author response to R4**

We thank the reviewer for their comments which have helped to strengthen this manuscript. Please find our responses below:

1) The reviewer shared the concerns of others with respect to the use of the CCI data and performance metrics used within the paper.

We have included extra text explaining the limitations of the used observations and have also included unbiased RMSD and correlation statistics for our experiments to show that the DA is not just acting to match the climatology of GLDAS1-Noah soil moisture. This is the same text from our response to reviewer 3's second comment, P11L14:

"Satellite soil moisture products can be subject to larger errors and biases associated with data processing. This is particularly true for the CCI level 3 combined active and passive product used in this paper, as in order to merge information from 11 different sensors data is CDF matched to the GLDAS-Noah v1 model (Rodell et al., 2004). Therefore, any bias within the GLDAS-Noah model will be included in the level 3 soil moisture product used here. To make sure we are not just correcting the bias of the JULES model to that of GLDAS-Noah we include summary statistics of unbiased root-mean squared difference (ubRMSD) and temporal correlation in table 2. In every case we find that after data assimilation we improve both ubRMSD and correlation and in the majority of cases find the best results for experiment 4 (TAMSAT v3.0 with DA). For the north of Ghana, we reduce the ubRMSD by 18% from experiment 3 (0.0622 m3 m−3) to experiment 4 (0.0508 m3 m−3). From experiment 2 to 4 we can see that, after data assimilation, using TAMSAT v3.0 rainfall over v2.0 has contributed to a 6% reduction in ubRMSD when calculating statistics over the

whole period. In the south of Ghana, we reduce the ubRMSD by 21% from experiment 3 (0.0590 m3 m−3) to experiment 4 (0.0467 m3 m−3), here improved rainfall data has contributed to 10% of this reduction. We find the highest correlations in the north of Ghana for the whole period (2010 - 2014), this is mainly due to the seasonal cycle being much more pronounced in this region."

2) The reviewer asked us to explain the larger improvement and lower overall errors shown for the South in Figure 5, given that we mention the products have lower skill in the south.

In Figure 5 we are comparing our model estimates to the ESA-CCI observations which we believe will also have higher errors in the south given the denser vegetation cover and higher cloud cover. Therefore, lower errors in Figure 5 do not necessarily mean that our model estimate is closer to the truth in the south if, as expected, the CCI observations are also of poorer quality in this region. There is also a much more pronounced seasonal cycle in the north, so that the model mistiming this cycle slightly can lead to large peaks in error. We have since re-run our experiments with a 5 cm soil depth and find similar results. The new figures relating to this point are 7 and 8. We have updated the text to clarify this point. P9L3:

"Experiments 2 and 4 have a lower RMSD in the south (Figure 8) compared to the north (Figure 7), this seems surprising given that we consider the quality of the data to be poorer in the south. However, this is in part due to the much more pronounced seasonal cycle in the north leading to peaks in RMSD when the seasonal cycle is even slightly mistimed by the model. We also have less confidence in the CCI soil moisture observations in the south so a lower RMSD in comparison to this product over this region is perhaps not indicative of a better soil moisture estimate overall."

3) The reviewer asked us to indicate the version of the ESA CCI soil moisture we are using.

We are using ESA CCI level 3 v03.2. We have included this in the manuscript in section 2.3.

**Author response to SC1**

We thank the author for their short comment which has helped to strengthen our manuscript. Please find our response below:

1) The author asked if the CCI product used is CDF matched to the GLDAS- NOAH model and if so requested that we include different performance metrics.

We are using the level 3 combined product which is CDF matched to the GLDAS- NOAH model, we agree that including different performance metrics is best to show the reduction in random errors. We have included ubRMSD and temporal correlation as suggested.

2) The author comments that it would be very desirable to repeat this experiment over a data rich area in order to have much better ground based verification.

We agree this would be very beneficial to look at and we would be excited to talk to the author more about possibilities for the future. The work in this paper formed part of the ERADACS project which was based in Ghana, it was therefore necessary that the focus of the paper was over this region. Please do contact me for further discussion: e.pinnington@reading.ac.uk